# Engineering Hyaluronic Acid for the Development of New Treatment Strategies for Osteoarthritis

**DOI:** 10.3390/ijms23158662

**Published:** 2022-08-04

**Authors:** Yu Seon Kim, Farshid Guilak

**Affiliations:** 1Department of Orthopaedic Surgery, Washington University School of Medicine, St. Louis, MO 63110, USA; 2Shriners Hospitals for Children—Saint Louis, St. Louis, MO 63110, USA; 3Center of Regenerative Medicine, Washington University School of Medicine, St. Louis, MO 63110, USA; 4Department of Biomedical Engineering, Washington University, St. Louis, MO 63105, USA

**Keywords:** proteoglycan, aggrecan, regenerative medicine, arthritis

## Abstract

Osteoarthritis (OA) is a degenerative joint disease that is characterized by inflammation of the joints, degradation of cartilage, and the remodeling of other joint tissues. Due to the absence of disease-modifying drugs for OA, current clinical treatment options are often only effective at slowing down disease progression and focus mainly on pain management. The field of tissue engineering has therefore been focusing on developing strategies that could be used not only to alleviate symptoms of OA but also to regenerate the damaged tissue. Hyaluronic acid (HA), an integral component of both the synovial fluid and articular cartilage, has gained widespread usage in developing hydrogels that deliver cells and biomolecules to the OA joint thanks to its biocompatibility and ability to support cell growth and the chondrogenic differentiation of encapsulated stem cells, providing binding sites for growth factors. Tissue-engineering strategies have further attempted to improve the role of HA as an OA therapeutic by developing diverse modified HA delivery platforms for enhanced joint retention and controlled drug release. This review summarizes recent advances in developing HA-based hydrogels for OA treatment and provides additional insights into how HA-based therapeutics could be further improved to maximize their potential as a viable treatment option for OA.

## 1. Introduction

Osteoarthritis (OA) is a chronic inflammatory disease that is characterized by the gradual degradation of the cartilage extracellular matrix (ECM), as well as by pain and physical challenges associated with the deterioration of joint health. Multiple pathologic changes of the cartilage and bone are associated with the progression of OA [1,2,3]. The articular cartilage undergoes degradation, which is denoted by alterations in the mechanical properties of the matrix and the development of fibrillation and fissures of the cartilage [2]. Simultaneously, bone turnover is increased, leading to subchondral bone thickening and osteophytes formation on the joint margins, all of which signify aberrant bony remodeling in the joint [3]. Recently published data on OA prevalence indicate that as of 2019, there were more than 500 million OA cases globally [4]. With a lack of reliable treatment options, patients with OA suffer chronically from pain and loss of mobility, leading to a significant financial burden on the medical system, as well as on the individuals, due to the disease [5]. Most treatment options for patients suffering from OA focus on symptomatic relief and palliative care, such as physical therapy with anti-inflammatory medications [6,7].

While extensive research has been conducted to discover treatment options for OA, currently, the only reliable surgical option for OA is total arthroplasty or joint replacement [8,9]. However, joint replacement surgeries can be costly and require extensive rehabilitation efforts from the patient. Most importantly, artificial joints are not a permanent solution and generally require a replacement surgery a decade after the initial surgery, thus making the joint replacement surgery a non-viable option for young and middle-aged patients with severe OA. This void in viable treatment options for OA has primed the field of orthopedics and tissue engineering to devise novel methods to not only alleviate the pain associated with OA but also to halt its progression and regenerate the damaged cartilage tissue. Specifically, tissue engineering aims to design constructs out of a specific combination of cells, drugs, and scaffolds that can then be delivered to the damaged tissue and encourage its regeneration. For constructs that specifically target OA cartilage, various combinations of anti-inflammatory drugs, cells, and scaffold/hydrogel materials that mimic the native cartilage ECM have been investigated [10,11,12].

Multiple biological polymers such as alginate [13], gelatin [14], and chondroitin sulfate [15], as well as synthetic polymers, such as poly(ε-caprolactone) (PCL) [16] and poly(lactic-co-glycolic) acid [17,18], have been successfully utilized to develop tissue-engineered constructs for treating OA-affected joints. In particular, one of the materials that has been gaining increasing attention for developing tissue-engineered constructs for OA treatment is hyaluronic acid (HA), a non-sulfated glycosaminoglycan composed of a repeating disaccharide unit of D-glucuronic acid and N-acetyl-D-glucosamine. HA can be isolated directly from animal tissues or produced by genetically-modified bacteria and microorganisms [19,20]. HA from different sources vary in their average molecular weight (MW) and distribution, which in turn determines other physicochemical properties, such as the degradation profile of HA [21] and the stiffness of the HA-derived tissue-engineered constructs [22].

HA is present in both the synovial fluid and as a component of the cartilage ECM in the joints and has demonstrated various therapeutic effects for the treatment of OA, which are discussed in-depth in multiple review articles [23,24]. One of the most significant roles played by HA is its physicochemical role in supporting joint lubrication, which has led to the development of various injectable HA-based therapeutics in an attempt to restore the degraded viscoelastic properties of the synovial fluid [23,24,25,26]. However, the long-term benefits of HA injections are still debated, as they have been shown to only provide temporary relief from OA-related pain and do not slow down the progression of the disease nor treat the damaged tissue [27,28,29]. Moreover, meta-analyses of randomized controlled trials that compare HA injections to placebos suggested that HA injections are not only clinically ineffective compared with placebo but also may carry a greater risk of adverse events such as inflammatory response following the treatment [30,31,32].

Compared with other biomacromolecules used to develop tissue-engineered constructs for OA treatment, HA has the potential biologic advantage of being a major component of synovial fluid and the cartilage ECM. Extensive research, therefore, has focused on developing tissue-engineered HA-based therapies that improve upon the biocompatibility and inherent biological and physicochemical roles played by HA in the joints. While various publications have summarized the current status of utilizing HA to develop tissue-engineered constructs for bone and cartilage tissue regeneration [33,34], such efforts have not recently been made for applications that pertain to OA treatment. This short review thus aims to summarize recent advances in developing HA-based hydrogels to improve upon the current injection method for joint lubrication and combining the therapeutic benefits of HA with those of other components, such as stem cells and drug molecules, for treating OA-affected joints. For a summary of recent progress in fabricating HA-based constructs for general tissue-engineering applications, readers are directed to the following review articles [35,36,37]. To keep a clear distinction between HA hydrogels developed for cartilage regeneration versus those for OA-affected joint treatment, reports published within the last four years that specifically discuss OA-targeted applications were reviewed. By providing an overview of these recent innovations, this review intends to shed light on potential future improvements in the treatment of OA.

## 2. HA Hydrogels for Joint Lubrication

As one of the principal components of articular cartilage ECM, HA functions as a framework to which aggrecan can covalently bind, thus forming a proteoglycan (PG) aggregate that plays an integral role in providing compressive load resistance to articular cartilage [38]. HA is also found as a component of synovial fluid, where it is the main regulator of fluid viscoelasticity and provides lubrication to the joint [23]. During the early progression of OA, the type II collagen network undergoes proteolytic degradation by members of the matrix metalloproteases (MMPs), which leads to the release of PG aggregates embedded within the cartilage matrix [23,39]. Further enzymatic degradation of PG core proteins exposes HA to the inflammatory conditions in the OA joint and results in its depolymerization into low-MW fragments [40,41,42]. These characteristics highlight the potential of replenishing high-MW HA as a potential therapeutic approach for the management of OA progression.

One of the potential reasons for the reported ineffectiveness of HA injection treatment is the short retention period of HA within the joint cavity, thus requiring repeated intra-articular injections to demonstrate its therapeutic efficacy [43,44]. Several research studies have thus focused on enhancing the retention of HA in the synovial fluid by delivering HA as a component of a composite hydrogel [45,46,47]. The fabrication of such hydrogels most often involves modifying the backbone of HA with a reactive group, such as tyramine [46], vinyl sulfone [47], and hydrazide [48], and combining it with a second polymeric chain that will undergo either a non-covalent interaction or covalent crosslinking with the modified HA chain. The second chain could be a long polymer or biomacromolecule, or a short crosslinker that enables HA chains to form a tight network. The resulting hydrogel network, when compared with HA delivered as a solution, will be less susceptible to enzymatic degradation and thus provide a prolonged HA-release profile. This strategy of fabricating HA-based composite hydrogel has recently been explored in clinics. RegenoGel is an injectable, covalently conjugated HA–fibrinogen hydrogel that undergoes gelation within the joint by the interaction of thrombin on fibrinogen. Preliminary results from clinical trials have indicated that RegenoGel can significantly improve pain outcomes compared with placebo controls while having good safety outcomes [49]. While the product is still undergoing clinical trials, such promising results could propel the development and approval of more HA-based composite hydrogels for joint lubrication. HA retention can also be enhanced by functionalizing HA with inhibitors for hyaluronidase and other proteases. For instance, HA covalently modified with a small-molecule MMP inhibitor showed better resistance to degradation by hyaluronidase compared with unmodified HA while demonstrating similar viscoelastic properties to those of human synovial fluid [50].

In addition to its relatively short retention time post-injection, another reason HA injection into the synovial joint is not considered to be a clinically effective treatment method is because HA, when simply injected, fails to localize to the joint surface. It has been demonstrated that while HA supplementation will enhance the viscoelasticity of the synovial fluid, lubrication of the cartilage surface is not achieved if HA is not localized to the surface [51] to interact with lubricin (PRG4) to form a lubricating boundary layer [52]. One approach involves conjugating tissue-adhesive groups onto the backbone of HA to enhance HA retention on the joint surface. One such example is dopamine-conjugated HA, which has been researched for its bio-adhesive properties and has seen its application as a tissue adhesive and anti-fouling coating on biological devices [53,54,55]. The usage of HA–dopamine as an injectable cartilage lubricant was investigated by using an ex vivo bovine cartilage model, which showed that compared with non-modified HA, HA–dopamine could better adsorb onto the cartilage surface and thus provide better boundary lubrication [56]. Another method of improving HA localization involves using an intermediate adhesive that can guide injected HA to the cartilage surface. For instance, a heterobifunctional poly(ethylene glycol) (PEG) conjugated with HA-binding and collagen-binding peptide could act as a glue on the joint surface and assist with retaining injected HA [57].

HA hydrogel can also be used in conjunction with other molecules and particles that can act as lubricants. Liposomes are one such class of molecules that have been identified to lower the friction coefficient of a biological surface via boundary lubrication [58,59,60]. The combined lubrication effects of liposomes with HA hydrogel have thus been explored as a potential treatment option for OA-affected joints [61,62,63]. This strategy allows encapsulated liposomes to provide enhanced boundary lubrication to HA hydrogel by forming a hydration shell on the gel surface and preventing the loss of water molecules from the bulk hydrogel. The synergistic effect of the HA hydrogel-mediated delivery of liposomes was demonstrated using a tendon explant model, where the co-delivery method significantly reduced the friction coefficient of the tissue surface compared with HA- or liposome-only delivery [63]. Boundary lubrication could be further improved by conjugating dopamine to the backbone of HA to further enhance its localization and adhesion to the tissue surface. Another consideration for liposomes to form an effective hydration shell on the gel surface is to ensure the proper rearrangement of liposomes from the bulk hydrogel to the surface. This provides a challenge for hydrogels formed via covalent crosslinking, as the liposomes in the bulk phase are trapped by the tight hydrogel network and thus cannot provide boundary lubrication. One approach to tackle this issue involved the formation of a self-healing, shear-responsive, and covalently crosslinked HA hydrogel by mixing aldehyde- and hydrazide-modified HA [61]. The dynamic covalent-linked hydrogel could restructure in response to shear and allow encapsulated liposomes incorporated in the hydrogel to rearrange to the surface layer and provide lubrication to the joint.

Delivering HA as micro- and nanogels is another approach to enhance HA retention and joint lubrication [64,65,66,67]. For instance, injectable and thermoresponsive HA nanogels can be fabricated by conjugating poly(N-isopropylacrylamide) (PNIPAM) chains onto the backbone of HA [64,66]. PNIPAM is a polymer that demonstrates thermoreversible properties, where it undergoes a phase transition from solution to gel as the temperature is increased above its lower critical solution temperature (LCST) of around 32 °C [68], which is tunable by modulating the hydrophobicity and hydrophilicity of the copolymer [69,70]. PNIPAM-grafted HA, once injected into the body, spontaneously formed into nanogels, which demonstrated enhanced retention and resistance to proteolytic degradation, as well as its better protection against OA-induced GAG loss compared with HA injections [64]. The surface modification of HA microgels can further enhance the natural lubrication properties of HA [71]. Significant reductions in friction coefficient were observed from HA microgels grafted with 2-methylacryloyloxyethyl phosphorylcholine (MPC) compared with unmodified microgels. When injected into the joints of rats with early-stage OA, MPC-grafted HA microgels enhanced the expression of aggrecan compared with unmodified HA microgels. The therapeutic effect of MPC-grafted HA microgels could be further enhanced by loading the microgels with diclofenac sodium, an anti-inflammatory drug [71].

In summary, various strategies have been developed that aim to enhance the role of HA as a joint lubricant, although their clinical efficacy remains to be shown. These techniques range from delivering HA as part of a crosslinked composite hydrogel to forming HA in microgels. Multiple design factors have been incorporated to provide additional properties to the HA hydrogel, such as incorporating tissue-adhesive moieties onto the backbone of HA for joint surface localization and loading liposomes for surface hydration and lubrication. While designing HA hydrogels to enhance their inherent therapeutic properties of joint lubrication has proven to be beneficial, the next two sections will illustrate the further possibilities of using HA to design hydrogels for biologics and cell delivery.

## 3. HA-Based Hydrogels for Biologics Delivery

While the direct intra-articular delivery of drug molecules has shown promising results with symptomatic relief, such an approach requires a recurring injection due to the short half-life of the drugs within the joint space [72]. Tissue-engineered drug delivery platforms are designed to allow for a longer retention and release profile of the drug molecules, thereby reducing the need for recurring injections while preventing any toxicities that could arise from the bulk release of drugs. The repeating disaccharide unit of HA contains carboxylic, hydroxyl, and acetyl groups, which can be modified using various bioconjugation techniques to covalently tether biological molecules.

Successful applications of HA-based hydrogels as delivery systems for various anti-inflammatory compounds, such as epigallocatechin-3-gallate (EGCG) [73], diclofenac potassium [74], and sulforaphane (SFN) [75], have been reported. A controlled delivery of EGCG, a polyphenolic compound with anti-oxidant and anti-inflammatory properties, was achieved by covalently conjugating the molecule onto the backbone of HA delivering it as part of a bulk HA/gelatin hybrid hydrogel. The gel demonstrated enhanced chondroprotective properties in a mouse OA model, as demonstrated by significantly thicker cartilage compared with either direct injection of EGCG or unloaded hydrogel [73]. For a more sustained drug release, drug molecules can also be loaded into particle-based carriers, which are then encapsulated in HA hydrogel for delivery. For instance, HA hydrogel loaded with diclofenac- and dexamethasone-encapsulated liposomes demonstrated anti-inflammatory effects on the knee joints of OA mice over a four-week period [76]. In addition to chemical compounds, HA-based hydrogels have also been used for the delivery of biological and cell-derived factors [77,78]. Thiol-modified HA and methacrylate-modified poloxamer 407 were used to fabricate an injectable biosynthetic hydrogel that could be used for the intra-articular delivery of keratinocyte growth factor 2 (KGF-2), which has been shown to demonstrate anti-inflammatory effects [77]. Compared with KGF-2 in solution, hydrogel-loaded KGF-2 delivered to OA-induced rat joints resulted in lower GAG loss and an enhanced tissue morphology from OA-induced inflammation. A similar strategy was used to enhance the therapeutic efficacy of mesenchymal stem cell (MSC)-derived extracellular vesicles (EVs) [78]. Encapsulating EVs in HA/PEG composite hydrogel allowed for a more controlled release platform for EVs, as well as providing protection against proteases in an in vivo rat OA model.

Stimuli-responsive HA hydrogels can adjust the drug release profile according to the joint environment and thus act as an on-demand drug delivery system. For instance, an injectable, supramolecular hydrogel that consists of a tightly packed nanoparticle was formed by the physical blending of carboxymethyl hexanoyl chitosan with HA [79]. The hydrogel demonstrated dynamic changes in porosity with respect to pH, where lower pH resulted in lower porosity and thus a more sustained release of drugs from the bulk hydrogel phase. Higher pH led to higher porosity, resulting in a faster degradation profile. As the joint environment during OA experiences a reduction in pH, this pH-dependent drug release and degradation profile can be adopted to design stimuli-responsive hydrogels that provide a sustained release of drugs to an inflamed joint, and which subsequently undergo rapid degradation once the joint gets recovered.

HA-based microgels have been proven to be excellent candidates for developing drug delivery strategies for OA treatment, as the delivery platform can combine the inherent biological properties of HA along with the therapeutic efficacy of encapsulated drug molecules. This approach also bypasses the issues of biocompatibility and potential cytotoxicity in the use of synthetic-polymer-based microcarriers. Indeed, the successful encapsulation and delivery of multiple anti-inflammatory drugs such as curcuminoid [80] and diacerein [81], as well as growth factors [82], have been reported for OA treatment applications. Using a microfluidics device, HA- and heparin-methacrylate were covalently crosslinked to fabricate porous microgels, which were then loaded with PDGF-BB and TGF-β3 [82]. PDGF-BB functioned as a stem-cell-recruiting factor, while TGF-β3 guided the chondrogenic differentiation of recruited stem cells and HA acted as a substrate onto which the recruited stem cells could adhere to. Rat OA joints treated with growth-factor-loaded microgels showed better tissue regeneration compared with those treated with either unloaded microgels or growth factor delivered in a solution. A similar microfluidics-based strategy was used to fabricate photo-crosslinked HA microgels encapsulated with rapamycin-loaded liposomes [67]. Friction between the joints allowed liposomes to rearrange to the surface of the microgels and provide joint surface lubrication, which in turn allowed for a sustained release of rapamycin from the liposomes. Rapamycin has been shown to have a chondroprotective effect by activating autophagy via mTOR pathway inhibition [83,84], and its therapeutic effects could be improved by allowing for a more sustained, localized delivery using the liposome-encapsulated HA microgel platform.

The fabrication of HA-based hydrogels that rely on the inherent therapeutic effects of HA have also been reported. One such system was designed as a composite hydrogel formed via Michael addition reaction between HA and a synthetic triblock copolymer [85]. HA released from the hydrogel as the gel underwent bulk degradation could not only downregulate the expression of pro-inflammatory cytokines, but also induce an anabolic response from bone marrow MSCs in vivo in a mouse OA model. Another HA delivery system was achieved by modifying 5β-cholanic acid onto the backbone of HA and allowing the chains to form self-assembled HA nanoparticles via hydrophobic interaction [86]. The nanoparticles demonstrated high retention within the joint and resistance to proteolytic degradation, and could directly interact with CD44 clusters on cell membranes. These interactions inhibited the binding between CD44 and the fragmented, low-MW HA present in the synovial fluid of OA joints, which has been shown to trigger an increase in the expression of catabolic genes and pro-inflammatory cytokines [23,87].

In addition to the delivery of biomolecules and growth factors, HA-based hydrogels for gene delivery have been developed for silencing the expression of enzymes that are linked with OA progression [88,89,90]. For example, ADAMTS-4 and ADAMTS-5, members of the A Disintegrin and Metallo-Proteinase with Thrombospondin Motifs (ADAMTS) family, have been shown to degrade proteoglycans and contribute to early OA development [91,92,93]. Knocking out either or both enzymes could successfully protect cartilage from proteoglycan degradation and thus decrease the degree of OA progression, both in vitro and in vivo [94,95]. A hydrogel-based gene-silencing method was subsequently developed, where locked nucleic-acid-modified antisense nucleotides designed to arrest the translation of ADAMTS-5 mRNA were loaded in fibrin-HA hydrogel and used to transfect hydrogel-loaded chondrocytes in vitro [88]. ADAMTS-5 expression was knocked down for up to 14 days, which highlights hydrogel-mediated gene silencing as a potential therapeutic approach for OA treatment. An on-demand gene-silencing method using gold nanorods and a gapmer antisense oligonucleotide delivery system has also been reported [90]. This approach first involved fabricating spherical nucleic acids (SNAs) by modifying the antisense DNA sequence of IL-1β mRNA onto gold nanorods, which were then conjugated to the backbone of complementary oligonucleotide-grafted HA hydrogel via DNA hybridization. The release of SNA from the hydrogel could be controlled in vivo by using near-infrared light to induce photo-thermal DNA dehybridization.

Overall, HA-based hydrogel has proved to be an effective, versatile platform for delivering various therapeutic compounds, ranging from anti-inflammatory drugs to antisense nucleotides, to OA-affected joints. These strategies benefited from the biocompatibility of HA as well as its versatility in being easily modifiable with various functional groups to develop HA-based hydrogel drug delivery methods with enhanced drug retention and release profiles. In addition, HA hydrogels that have inherent therapeutic effects have been developed, which further emphasize the synergistic effect that can be achieved from using HA hydrogels as a drug delivery platform.

## 4. HA-Based Hydrogels for Cell Delivery

In addition to its physicochemical roles, HA also provides biological cues to cells by triggering cell proliferation [96], stimulating anabolic responses [97,98], and controlling the expression of inflammatory cytokines and proteases [99,100]. HA also has binding sites for several different cell surface receptors, such as CD44 and ICAM-1 [101,102]. In particular, CD44-mediated binding between HA and chondrocytes plays an important role in the retention of aggrecan aggregate to cartilage ECM, thus preventing the loss of proteoglycan and, eventually, the degradation of cartilage ECM [103]. These properties, in addition to its inherent biocompatibility, have allowed HA to be successfully used in developing cell-laden HA hydrogels for various tissue-engineering applications. Their efficacy in regenerating OA joints has also been recently demonstrated, and in some cases cell-laden HA hydrogels have been shown to be superior to HA injection in reducing the pain levels of patients [104]. Such results highlight the potential of cell delivery as an effective OA therapy. Efforts have thus been made to develop HA-based hydrogels that could deliver cells to the OA joint, thereby harnessing the benefits of both components.

One such approach involves the co-delivery of stem cells with therapeutic compounds in HA-based hydrogels [105,106]. For instance, bone marrow MSCs were encapsulated in HA–Poloxamer 407 composite hydrogel along with icariin, the main active component of the herb *Epimedium*, which has been shown to be chondroprotective in addition to be able to guide MSC chondrogenesis [107,108,109]. HA-based composite hydrogels allowed the solubilization and sustained delivery of icariin due to hydrogen bonding between HA and the compound. In a destabilization of the medial meniscus (DMM)-induced OA model, hydrogel loaded with both MSCs and icariin showed significantly lower OARSI and Mankin scores compared with both cell-only and icariin-only groups, demonstrating the synergistic effect of co-delivering cells with therapeutic compounds for the treatment of OA. Another approach involved encapsulating cell spheroids instead of cell suspension [106]. MSC spheroids were formed with kartogenin-loaded short electrospun polylactide–PEG fibers to aid chondrogenic differentiation, before being loaded into an injectable PEG-HA hydrogel containing celecoxib-loaded short fibers for anti-inflammatory effects. In a rabbit OA model, joints treated with spheroid-encapsulated constructs showed better protection of the cartilage tissue compared with the control group.

While multipotent adult stem cells can be isolated from various tissues, such as adipose tissue and bone marrow, their numbers are limited, and they have limited expansion capacity in vitro before losing their multipotency. As pluripotent stem cells (PSCs) have near unlimited capacity to expand without losing their differentiation capabilities, PSC-derived progenitors can be a practical alternative to MSCs. The synergistic effects of PSC-derived progenitors and HA hydrogels were demonstrated recently using a monoiodoacetate-induced rabbit OA model, where embryonic stem cell (ESC)-derived progenitors encapsulated in HA hydrogels outperformed both the acellular HA hydrogel and the cells delivered as a cell suspension [110].

Hydrogel-encapsulated cells can also act as a resident producer of disease-modifying drugs. For this purpose, cells are first genetically modified to produce or overexpress proteins that can protect cartilage from OA before being encapsulated in a hydrogel and delivered to the intra-articular region. For instance, HA–collagen hydrogel was successfully used to deliver genetically modified adipose derived stem cells that overexpress TGF-β1, a paracrine growth factor with anti-inflammatory effects [111,112]. The cell-laden hydrogel demonstrated chondroprotective and anti-inflammatory effects in a rat OA model, as evidenced by reduced TNF-α concentration and a lower synovitis score compared with the no-treatment group [113].

HA-based hydrogels have also been used as bioinks in conjunction with other polymeric bioinks to fabricate 3D-printed cell-laden multiphasic scaffolds [114]. In this approach, HA bioink is used as a cell carrier and is usually co-printed with a second, stiffer bioink that can provide structural support for the cell-laden HA bioink. For instance, a 3D-printed composite scaffold designed for cartilage-layer regeneration was fabricated by printing MSC-encapsulated methacrylated HA (MeHA) bioink with a second PCL ink loaded with kartogenin, a small molecule that has been shown to stimulate chondrogenesis [114]. This layer was combined with a subchondral bone layer printed from β-tricalcium phosphate (β-TCP)-loaded PCL ink and an anti-inflammatory layer consisting of diclofenac sodium-loaded MeHA hydrogel to generate a multilayered 3D-printed construct for regenerating osteochondral defects in OA joints. Constructs implanted into the osteochondral defects of rats with medial meniscectomy-induced OA could successfully induce tissue repair while attenuating a joint inflammatory response. HA-based bioinks can be further modified to provide additional therapeutic benefits for applications in degenerative joint diseases while supporting the viability of encapsulated cells [115]. For instance, bioink that consists of HA modified with phenylboronic acid and gelatin demonstrated inherent antioxidant properties by boronate ester bonds [115]. The bioink effectively scavenged H_2_O_2_ and supported the viability of encapsulated chondrocytes.

No HA-based cell delivery platforms have yet been approved by the FDA for the treatment of knee OA. Their clinical benefits, however, have recently been demonstrated outside of the U.S. One such example is CARTISTEM, a therapy that combines allogeneic umbilical-cord-blood-derived MSCs with HA, which are mixed and implanted as a cell-encapsulated hydrogel at the defect site. The product acquired market approval by the Ministry of Food and Drug Safety in South Korea in 2012 and has demonstrated clinical efficacy among patients with knee OA [116,117]. In a two-year follow-up study, 128 patients with knee OA who were treated with CARTISTEM showed significant improvements in clinical outcomes compared with their pre-operative conditions. While longer-term clinical follow-up studies have not been published so far, these results emphasize the potential of using stem-cell-encapsulated HA hydrogel as a viable treatment option for OA patients.

## 5. Concluding Remarks

As a chronic degenerative and inflammatory disease, OA affects the lives of hundreds of millions of people around the globe. HA injection is one of the most widely used methods to treat OA pain; however, its efficacy in altering disease progression has not been shown. In this review, we have summarized recent advances in fabricating HA-based hydrogel therapeutics that are designed to improve the inherent biological effects of HA, as well as those that rely on its biocompatibility to design novel cell and drug delivery systems for OA (Figure 1). These hydrogels have demonstrated abilities to not only enhance the retention and localization of HA on the joint surface for enhanced lubrication but also deliver cells and biologics to impede the progression of OA and regenerate the damaged tissue. While the results of these studies point towards a potential therapeutic benefit of HA hydrogels for the treatment of OA, there are several considerations that need to be addressed for HA hydrogels to show efficacy in this regard.

While the biological effects of HA are well-documented and have led to the successful development of non-drug-loaded HA hydrogels with inherent therapeutic effects [85,86], HA is often considered simply as one of the structural components of the bulk hydrogel and not necessarily as a bioactive molecule. For instance, while the MW of HA is an important factor in deciding its therapeutic efficacy once injected into the joint [118,119,120], it was rarely selected as an experimental variable in fabricating and validating the therapeutic efficiency of HA hydrogel [121]. Focusing on developing HA-based hydrogels that capture not only the therapeutic benefits of the encapsulated drugs and biomolecules but also those of HA will result in a tissue-engineered construct that captures the benefits of both components.

The backbone modification of HA is oftentimes necessary to form composite hydrogels that are not only more resistant to proteolytic degradation than HA delivered as a solution but also able to encapsulate and deliver cells and drug molecules to the joint. However, conjugating functional groups on HA can negatively impact its inherent biological properties. For instance, cells interact with HA via their cell surface receptor CD44, and reports have shown that the interaction between CD44 and HA can suppress the progression of OA [122]. However, introducing chemical groups on the repeating units of HA reduced not only the extent of binding between HA and CD44 but also the degree of chondrogenesis observed from encapsulated MSCs [123]. As the chemical modification of HA is oftentimes inevitable for fabricating hydrogels with appropriate physical properties for drug/cell delivery or joint lubrication applications, such drawbacks must be carefully considered to maximize the therapeutic benefits of HA-based hydrogels.

While MSCs are one of the most widely used cell sources for developing cell-encapsulated tissue-engineered constructs for bone and cartilage regeneration, their limited availability and minimal in vitro expansion capacity remain the two largest practical barriers to clinical applications. In this regard, induced pluripotent stem cells (iPSCs) provide an excellent alternative to MSCs as a cell source for OA treatment and cartilage regeneration. Indeed, an efficient protocol for differentiating iPSCs to MSCs has been established following a small molecule screening with iPSCs to select for specific combinations that result in cells with the highest expressions for MSC markers (CD73 and CD105) [110]. The usage of iPSCs with HA hydrogels has also been successfully demonstrated in other tissue types [124,125], further highlighting the potential of developing iPSC-encapsulated HA hydrogels for OA. Nevertheless, safety concerns regarding the usage of iPSCs must be thoroughly evaluated before their usage can be translated into clinics. One such concern is the complex lineage differentiation regimen and generation of off-target cells during the differentiation process. There have thus been ongoing efforts to design optimized differentiation protocols for obtaining chondrocytes from iPSCs; for instance, by sorting differentiated cells for COL2A1, which is a widely used marker for chondrocytes [126], or by using a mixture of small molecules and inhibitors to limit off-target differentiation during chondrogenesis [127]. Further optimization of iPSC differentiation pathways will eventually lead to their more widespread usage in developing iPSC-based therapies for OA treatment.

Along with the biological properties of HA and the therapeutic effects of cells and drugs that are being delivered, the physicochemical properties of the HA hydrogel should be carefully considered when designing HA-based therapeutic constructs for OA. Specifically, mechanical properties such as crosslinking density, stiffness, and degradation rates must be carefully tested and optimized for constructs that require prolonged residency within the joint space for the sustained delivery of cells and drug molecules or that are designed for tissue repair/replacement. Various research has thus been conducted to investigate the relationship between the different crosslinking densities and degradation profiles of HA-based hydrogels [128,129,130]. These results could be used to guide the development of the constructs, with the goal of maximizing their therapeutic potential.

In conclusion, HA has proven itself to be a promising biomacromolecule that can be used to fabricate tissue-engineered hydrogel constructs that target OA-affected joints thanks to its biocompatibility, versatility in being able to be conjugated to a variety of other materials, and ability to support cell viability. HA hydrogels demonstrated enhanced abilities not only to enhance the viscoelastic properties of the synovial fluid but also to regenerate the degraded cartilage ECM. While these platforms are still far from being used in clinics, HA hydrogels will continue to be a powerful, versatile tool for discovering new treatment options for OA.

## Figures and Tables

**Figure 1 ijms-23-08662-f001:**
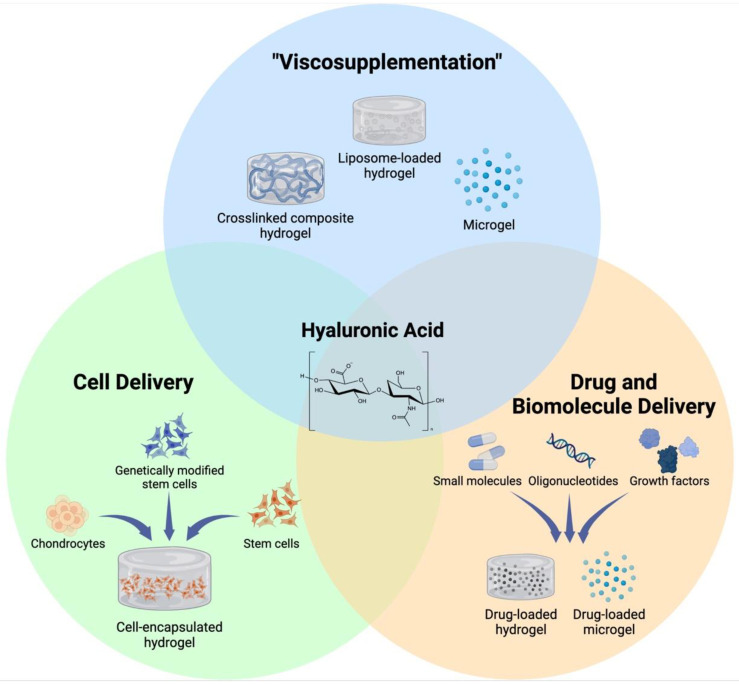
Different applications of HA-based hydrogels. Ease of chemical modification and excellent biocompatibility of HA allows for development of various HA-based hydrogels that are designed to serve distinct purposes for treatment of OA. Created with BioRender.com (accessed on 5 June 2022).

## Data Availability

The data presented in this study are openly available on PubMed.

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
