# Peer review of "Engineering Hyaluronic Acid for the Development of New Treatment Strategies for Osteoarthritis"

_ijms, 2022, doi:10.3390/ijms23158662_

Round 1
Reviewer 1 Report
The manuscript entitled ‘Engineering hyaluronic acid for the development of new treatment strategies for osteoarthritis’ is a very short review that presents important aspects related to the use of hyaluronic acid (HA) in different formulas, for the treatment of osteoarthritis (OA). The document is well written and concise, offering data about the binding and the biontegration of the HA in the host body. However, considering that it is a Review article, it should present some other important aspects that might shad light and offer complete information about this subject. The authors should describe the anatomy of the bones when they are affected by OA, which is going to be very useful for researchers working to produce novel HA treatments and implants. Different manufacturing techniques should be also mentioned, together with mechanical properties and degradation behaviors that have been investigated by now, by other authors which performed experimental research. In order this Review to be complete, it should go beyond some biology aspects and include the above-mentioned sections (chemical parameters that play key role in HA manufacturing, cross-linking, overall properties, limitations).
If this information is added, the manuscript can be published.
Regards,
Author Response
We thank the reviewer for the insightful comment and have added the following sentences to provide additional information on the subject.
Multiple pathologic changes of the cartilage and bone are associated with the progression of OA [1–3]. Articular cartilage undergoes degradation, which is denoted by alterations in the mechanical properties of the matrix and development of fibrillation and fissures of the cartilage [2]. Simultaneously, bone turnover is increased leading to subchondral bone thickening and osteophytes formation on the joint margins, all of which signify aberrant bony remodeling in the joint [3].(1. Introduction, paragraph 1)
HA can be isolated directly from animal tissues or produced by genetically-modified bacteria and microorganisms [19,20]. HA from different sources vary in their average molecular weight (MW) and distribution, which in turn determines other physicochemical properties such as degradation profile of HA [21], and stiffness of the HA-derived tissue engineered constructs [22]. (1. Introduction, paragraph 3)
Along with biological properties of HA and the therapeutic effects of cells and drugs that are being delivered, physicochemical properties of the HA hydrogel should be carefully considered when designing HA-based therapeutic constructs for OA. Specifically, mechanical properties such as crosslinking density, stiffness, and degradation rate must be carefully tested and optimized for constructs that require prolonged residency within the joint space for sustained delivery of cells and drug molecules or are designed for tissue repair/replacement. Various research has thus been conducted so far to investigate the relationship between different crosslinking densities and degradation profiles of HA-based hydrogels [128–130]. These results could be used to guide the development of the constructs with the goal of maximizing their therapeutic potential. (5. Conclusion, paragraph 5)
Reviewer 2 Report
In a manuscript entitled "Engineering hyaluronic acid for the development of new treatment strategies for osteoarthritis" by Yu Seon Kim and Farshid Guilak, the authors present the latest trends in osteoarthritis treatment. Although the research is still in the testing phase, the research to date is auspicious. The manuscript is well organized and supported by many references.
However, a few things need to be clarified.
Lines 95-97: Please describe how the molecular weight distribution of HA changes in a healthy and OA joint.
Line 169: What is the typical value of lower critical solution temperature?
Author Response
We thank the reviewer for the insightful comment and have added the following sentences to provide additional information on the subject.
During the early progression of OA, type II collagen network undergoes proteolytic degradation by members of the matrix metalloproteases (MMPs) and leads to the release of PG aggregates embedded within the cartilage matrix [23,39]. Further enzymatic degradation of PG core proteins exposes HA to the inflammatory conditions in the OA joint and results in its depolymerization into low-MW fragments [40–42]. These characteristics highlight the potential of replenishing high-MW HA as a potential therapeutic approach for the management of OA progression. (2. HA hydrogels for joint lubrication, paragraph 1)
Line 169: What is the typical value of lower critical solution temperature?
We thank the reviewer for the insightful comment and have added the following sentences to provide additional information on the subject.
PNIPAM is a polymer that demonstrates thermoreversible properties, where it undergoes phase transition from solution to gel as the temperature is increased above its lower critical solution temperature (LCST) of around 32 °C [68], which is tunable by modulating the hydrophobicity and hydrophilicity of the copolymer [69,70]. (2. HA hydrogels for joint lubrication, paragraph 5)